# Current Status of Recombinant Human Bone Morphogenetic Protein-2 (rhBMP-2) in Maxillofacial Surgery: Should It Be Continued?

**DOI:** 10.3390/bioengineering10091005

**Published:** 2023-08-24

**Authors:** Sung-Woon On, Sang-Yoon Park, Sang-Min Yi, In-Young Park, Soo-Hwan Byun, Byoung-Eun Yang

**Affiliations:** 1Division of Oral and Maxillofacial Surgery, Department of Dentistry, Dongtan Sacred Heart Hospital, Hallym University College of Medicine, Hwaseong 18450, Republic of Korea; drummer0908@hallym.or.kr; 2Department of Artificial Intelligence and Robotics in Dentistry, Graduated School of Clinical Dentistry, Hallym University, Chuncheon 24252, Republic of Korea; psypjy0112@naver.com (S.-Y.P.); queen21c@hallym.or.kr (S.-M.Y.); denti2875@hallym.or.kr (I.-Y.P.); purheit@daum.net (S.-H.B.); 3Institute of Clinical Dentistry, Hallym University, Chuncheon 24252, Republic of Korea; 4Department of Oral and Maxillofacial Surgery, Hallym University Sacred Heart Hospital, Anyang 14066, Republic of Korea; 5Dental Artificial Intelligence and Robotics R&D Center, Hallym University Sacred Heart Hospital, Anyang 14066, Republic of Korea; 6Department of Orthodontics, Hallym University Sacred Heart Hospital, Anyang 14066, Republic of Korea

**Keywords:** BMP, bone morphogenetic proteins, bone graft, maxillary sinus augmentation, alveolar ridge augmentation

## Abstract

Recombinant human bone morphogenetic protein-2 (rhBMP-2) has shown potential in maxillofacial surgery owing to its osteoinductive properties. However, concerns about its safety and high cost have limited its widespread use. This review presents the status of rhBMP-2 use in maxillofacial surgery, focusing on its clinical application, efficacy, safety, and limitations. Studies have demonstrated rhBMP-2’s potential to reduce donor site morbidity and increase bone height in sinus and ridge augmentation; however, it may not outperform autogenous bone grafts. In medication-related osteonecrosis of the jaw treatment, rhBMP-2 has been applied adjunctively with promising results, although its long-term safety requires further investigation. However, in maxillofacial trauma, its application is limited to the restoration of large defects. Safety concerns include postoperative edema and the theoretical risk of carcinogenesis. Although postoperative edema is manageable, the link between rhBMP-2 and cancer remains unclear. The limitations include the lack of an ideal carrier, the high cost of rhBMP-2, and the absence of an optimal dosing regimen. In conclusion, rhBMP-2 is a promising graft material for maxillofacial surgery. However, it has not yet become the gold standard owing to safety and cost concerns. Further research is required to establish long-term safety, optimize dosing, and develop better carriers.

## 1. Introduction

Bone morphogenetic proteins (BMP), first identified by Urist [1] in 1965, are parts of the superfamily of transforming growth factor β (TGF-β) and possess osteoinductive properties. Although approximately 20 BMPs have been identified, BMP-2 and BMP-7 exhibit the most potent osteoinductive ability in bone and cartilage formation [2]. Since the cloning of the BMP gene became possible in the 1990s and its mass production became feasible [3], recombinant human BMP-2 (rhBMP-2) has been produced for therapeutic purposes. RhBMP-2 is used for various applications in maxillofacial surgery, starting with spinal surgery in orthopedic surgery. As it overcomes the donor-site morbidity of autografts and has an osteoinductive ability absent in xenografts, it has aroused much interest from clinicians and has enabled various clinical trials. Currently, numerous studies are being conducted to maximize the advantages of rhBMP-2.

Despite these advantages, concerns about the safety of rhBMP-2 have not completely disappeared. As a side effect, there are not only minor problems such as postoperative edema, but also the possibility of carcinogenesis, and various issues have been raised regarding its safety [4,5]. However, in most cases, it is considered a problem that may occur in a dose-dependent manner, and the possibility of serious complications mentioned above occurring due to the dose for clinical application is assumed to be very low [6]. Particularly in maxillofacial surgery, where bone grafting is frequently performed, it is necessary to review the clinical applications and complications of rhBMP-2, which is like a double-edged sword. The aims of this review were to examine the current status of the clinical application and efficacy of rhBMP-2 in the field of maxillofacial surgery and to review its safety and limitations, along with a literature review.

## 2. History and Characteristics of rhBMP-2

### 2.1. History of rhBMP-2

Research was initially conducted using bovine BMP, and then molecular cloning of human BMP became possible [7]. The osteoinductive ability of rhBMP-2 in spinal fusion and nonunion was verified in animal models, followed by evaluation in human clinical trials. In 2002, the rhBMP-2 product (INFUSE^®^ Bone graft; Medtronic, Memphis, TN, USA) obtained Food and Drug Administration (FDA) approval for application in anterior lumbar interbody fusion [8]. In addition, it obtained FDA approval in 2004 for treating acute open tibial fractures in skeletally mature patients [9]. In the field of maxillofacial surgery, in 2007, the FDA approved the use of rhBMP-2 for maxillary sinus augmentation and alveolar ridge augmentation [10]. Since then, it has been actively applied clinically, including off-label use.

Another noteworthy feature in the history of rhBMP-2 is the development of *Escherichia-coli*-produced rhBMP-2 (erhBMP-2). The widely used rhBMP-2, including INFUSE^®^, was a Chinese hamster ovary (CHO)-derived rhBMP-2 using an engineered CHO cell line. However, owing to its high cost, erhBMP-2 was developed to produce rhBMP2 from BMP-gene-transfected *Escherichia coli*. It showed comparable osteoinductive ability and clinical efficacy at a lower cost than the CHO-derived rhBMP-2 [11,12,13,14]. As discussed later, the high price of rhBMP-2 is one of its significant limitations, and these trials have become a stepping stone in bringing rhBMP-2 closer to clinicians.

### 2.2. Structure and Osteoinduction Mechanism of rhBMP-2

BMPs have a heparin-binding site and a cysteine knot associated with six cysteine residues [15]. These sites are known to regulate the process of bone formation by interacting with the endogenous heparin/heparin sulfate on the cell surface or in the extracellular matrix [16,17]. BMP-2 is a homodimer of two BMP molecules via a disulfide bridge [18,19]. The dimeric nature of BMP-2 is essential for its biological activity, as the protein is inactivated when the disulfide bridge is broken.

Human BMP-2 has a molecular weight of 32 kDa and 114 amino acid residues [20]. BMP-2 has many hydrophobic patches on its surface, so it shows limited solubility at physiological pH, which is a consideration for its pharmacological activity [19]. In addition to its rapid clearance rate, one of the characteristics of BMPs is that they are pleiotropic proteins. Therefore, it is crucial to consider this when manufacturing BMP-2 products because they can affect other molecular pathways besides bone regeneration [21,22]. If BMP-2 spreads to adjacent tissues other than the bone tissue, it may cause unwanted effects such as ectopic bone formation, resorption of native bone, and soft tissue swelling [23]. This suggests that developing a scaffold or carrier that can release BMP-2 only to the desired site with an appropriate dose is essential.

The primary osteoinduction mechanism of BMP-2 is to differentiate mesenchymal stem cells (MSCs) into osteoblasts. By binding to types I and II serine/threonine kinase receptors, BMP-2 activates Smad and non-Smad signaling pathways and ultimately activates the osteogenic genes, including runt-related transcription factor 2 (RUNX2) and osterix (Osx) for the differentiation of MSCs into osteoblasts [24]. Differentiated osteoblasts form the bone matrix and secondarily deposit calcium phosphate by secreting alkaline phosphatase to form bone. The activated osteoblasts are then embedded in the formed bone and act as osteocytes responsible for the additional support and structure of the bone [25]. Owing to the role of BMP-2 in osteogenesis, rhBMP-2 has been produced and approved by the FDA and is currently being applied in clinical practice (Table 1).

## 3. Clinical Application and Efficacy of rhBMP-2 in Maxillofacial Surgery

### 3.1. Maxillary Sinus Augmentation

Boyne et al. [26,27] contributed significantly to applying rhBMP-2 in maxillary sinus augmentation. They started a clinical pilot study in 1997 [27] and performed the first randomized controlled trial (RCT) in 2005 [26], demonstrating de novo tissue growth using rhBMP-2. They used an absorbable collagen sponge (ACS) as a carrier of rhBMP-2 to perform sinus floor augmentation in the posterior maxilla of 12 patients and reported an average bone height increase of 8.51 mm without serious or unexpected adverse effects [27]. The dose of rhBMP-2 was approximately 0.43 mg/mL, considered too low in bone quality, as evaluated by core bone biopsy. Therefore, they conducted an RCT in which rhBMP-2 and ACS were administered for maxillary sinus augmentation at two doses (0.75 and 1.5 mg/mL) of rhBMP-2 [26]. It was reported that both doses of rhBMP-2 and ACS yielded similar amounts of bone as those of the bone graft group and showed a safe profile. Therefore, they concluded that their results provided a basis for applying rhBMP-2 and ACS at a dose of 1.50 mg/mL in future studies. Subsequently, as a pivotal study, a multicenter prospective clinical trial was conducted in which 1.50 mg/mL of rhBMP-2 and ACS were applied for sinus floor augmentation in 160 patients at 21 centers in the United States [28]. Patients to whom rhBMP-2 and ACS were applied acquired a similar average bone height to those who underwent autogenous bone graft (7.83 ± 3.52 mm vs. 9.46 ± 4.11 mm) and showed comparable success rates in implant placement (79% vs. 91%). However, the rhBMP-2/ACS group showed no adverse events, whereas the autogenous bone graft group showed graft harvest-related complications in 17% of the cases. It was concluded that using off-the-shelf rhBMP-2 to eliminate the need for bone harvesting and consequent donor-site morbidity strongly supports its use in sinus floor augmentation. Based on these studies’ results, the FDA has approved using rhBMP-2 for maxillary sinus floor augmentation.

However, not all studies have shown such positive outcomes in sinus floor augmentation. Kao et al. [29] added Bio-Oss^®^, an inorganic bovine bone xenograft, to rhBMP-2/ACS, and compared the amount of new bone formation on histological specimens with Bio-oss^®^ alone. As a result, new bone formation was less than that in patients treated with Bio-oss^®^ alone, and they reported that adding rhBMP-2/ACS to Bio-oss^®^ had a negative effect on bone formation during maxillary sinus augmentation. In addition, in a prospective RCT in which erhBMP-2 and biphasic calcium phosphate (BCP) carriers were used for maxillary sinus augmentation and compared with Bio-Oss^®^ only [30], no significant radiological and histometric differences in bone volume were found between the two groups. Although a slight difference between the two groups was observed in the healing pattern in histological findings, it was concluded that ErhBMP-2/BCP in sinus augmentation could not improve bone regeneration compared to the conventional method using Bio-Oss^®^. In a systematic review and meta-analysis on the use of rhBMP-2 in maxillary sinus augmentation and localized alveolar ridge augmentation [31], rhBMP-2 substantially increased bone height but did not promote as much bone formation as an autograft or allograft. To summarize the efficacy of rhBMP-2 in maxillary sinus augmentation, the use of rhBMP-2 can reduce donor-site morbidity and increase bone height but does not provide substantial clinical benefit compared to other bone graft materials. Further research is needed to evaluate its long-term clinical success.

### 3.2. Alveolar Ridge Augmentation

The first study to apply rhBMP-2 in humans for alveolar ridge augmentation was Howell et al.’s 1997 clinical trial [32]. They performed alveolar ridge augmentation using rhBMP-2/ACS in 6 of 12 patients and evaluated its safety and osteoinduction during a 24-month follow-up period. The first four months were used to evaluate the short-term safety and technical applicability of rhBMP-2, while the remaining 20 months were used to evaluate the long-term safety. They reported that rhBMP-2/ACS was tolerated locally and systemically with no serious short- or long-term adverse effects. In addition, they suggested that the application of rhBMP-2/ACS could be a safe and feasible method for achieving localized defect augmentation.

Following Howell et al.’s study [32], Jung et al. [33] conducted a study in 2003 in which rhBMP-2 was applied with Bio-oss^®^ in guided bone regeneration. The study included 11 partially edentulous patients who underwent 34 dental implants in different areas of the jaw. The bone defects were randomly assigned to either the test or control group. Both groups were augmented with a bone substitute (Bio-oss^®^) and a resorbable collagen membrane, whereas the test group received Bio-oss^®^ coated with rhBMP-2. Measurements were taken at baseline and after a healing period of approximately six months, and the results showed that the test group exhibited a significantly greater reduction in defect height than the control group. Histomorphometric analysis revealed higher bone density and maturation in the test group, with a greater proportion of bone substitute particles in direct contact with the newly formed bone. They suggested that adding rhBMP-2 can enhance bone regeneration and improve the outcomes of guided bone regeneration. Based on these studies, Fiorellini et al. [34] conducted a multicenter randomized study in which rhBMP-2 was applied for extraction socket augmentation and dental implant placement with ACS at two concentrations. Eighty patients with buccal wall defects in the maxillary teeth immediately following extraction were enrolled, and patients were divided into groups receiving 0.75 mg/mL or 1.50 mg/mL rhBMP-2/ACS, placebo (ACS alone) or no treatment. The results demonstrated that the group receiving 1.50 mg/mL rhBMP-2/ACS showed significantly greater bone augmentation than the control group. The bone adequacy for dental implant placement was twice as great in the rhBMP-2/ACS group as in the placebo and no-treatment groups.

Furthermore, no differences were observed in bone density on histology between the newly induced and native bones. The study demonstrates that rhBMP-2/ACS significantly affects the de novo formation of alveolar bone for dental implant placement. Based on the results of Howell et al. [32] and Fiorellini et al. [34], the FDA has approved the use of rhBMP-2 for localized alveolar ridge augmentation.

In addition to ACS and Bio-oss^®^, materials used with rhBMP-2 for alveolar ridge augmentation include β-tricalcium phosphate and hydroxyapatite (β-TCP/HA), and demineralized bone matrix (DBM). β-TCP/HA are synthetic osteoconductive materials that provide a scaffold for bone formation and help stabilize blood clots [35]. In particular, β-TCP can entrap growth factors through micropores and prolongs their activity [36]. Based on this point, a randomized clinical trial was conducted where β-TCP/HA was applied together with rhBMP-2 for alveolar bone regeneration [37]. Seventy-two patients were divided into two groups: the experimental group received ErhBMP-2 coated β-TCP/HA, and the control group received β-TCP/HA graft material alone immediately after tooth extraction. Their results showed that the experimental group had significantly less reduction in alveolar bone height than the control group, and no adverse effects on the graft material were observed during the study. It was concluded that ErhBMP-2 coated β-TCP/HA could be more effective for alveolar bone preservation than conventional β-TCP/HA alloplastic graft materials [37].

DBM is classified as an osteoinductive material [38] and is made by removing inorganic minerals from donated human bone while leaving only the organic collagen matrix [39]. However, as there is controversy regarding its osteoinductive potential, the clinical use of DBM to obtain the benefits of native BMP release is limited. Focusing on this point, Kim et al. [40] combined a DBM gel with rhBMP-2 and applied it for alveolar ridge preservation after tooth extraction. In their randomized controlled trial, 69 patients were randomly assigned to the test group (rhBMP-2/DBM) or the control group (DBM alone). The safety of rhBMP-2/DBM was confirmed, and no adverse events or significant changes were observed. Radiographic measurements of the alveolar bone height and width showed no significant differences between the test and control groups. Their results concluded that injectable rhBMP-2/DBM can be easily and safely used in clinical applications for alveolar ridge preservation [40].

In a systematic review and meta-analysis on the use of rhBMP-2 in localized alveolar ridge augmentation [31], the overall effect size of 0.56 strongly supports the use of BMP over the control group in studies on localized alveolar ridge augmentation, and it was concluded that rhBMP-2 significantly increases alveolar bone height. Nevertheless, it should be noted that there was no difference in efficacy when the autogenous bone was applied to the control group. To summarize the efficacy of rhBMP-2 in alveolar ridge augmentation, rhBMP-2 can reduce donor-site morbidity and increase bone height (Figure 1), but does not provide substantial clinical benefit compared to autogenous bone. Additional studies are needed to evaluate long-term clinical success and adverse events.

### 3.3. Alveolar Cleft Reconstruction

Alveolar cleft reconstruction is an essential procedure in patients with cleft lip and palate for several purposes, such as achieving the continuity of the maxillary arch and helping the eruption of the maxillary canines. Autologous iliac bone was traditionally considered the gold standard as a bone graft material for alveolar cleft reconstruction [41]. Autologous iliac bone has the advantages of an autograft and can provide abundant amounts; however, it has the major disadvantage of donor-site morbidity. On the other hand, as the advantages of rhBMP-2, which can reduce the operation time and hospitalization period and the absence of complications related to the donor site, have emerged, rhBMP-2 has begun to be considered as an alternative for the reconstruction of the alveolar cleft. A study that applied rhBMP-2 to repair the alveolar cleft for the first time was conducted by Chin et al. [42] in 2005. Their case series reported successful osseous unions in 49 sites where rhBMP-2/ACS was applied to 50 sites out of 43 patients, suggesting that rhBMP-2 could be a treatment option to avoid iliac bone harvesting. Since then, studies have attempted to use hyaluronan-based hydrogels [43] or DBM [44] as scaffolds in addition to ACS, the primary scaffold of rhBMP-2.

In a systematic review and meta-analysis on the use of rhBMP-2 in alveolar ridge cleft construction in 2020 [45], the rhBMP-2 group exhibited higher average bone volume formation (61.11% vs. 59.12%) than the control group (iliac crest cancellous bone). However, the control group showed a higher average bone height formation (75.4% vs. 61.5%) than the rhBMP-2 group. The meta-analysis suggested that rhBMP-2 treatment may benefit bone formation compared to iliac crest grafts; however, the evidence was of low certainty. Consequently, conducting controlled clinical trials with more patients is recommended to establish the use of rhBMP-2 for alveolar cleft treatment. Summarizing the studies on rhBMP-2 for alveolar cleft reconstruction, although rhBMP-2 is promising, further studies are required to verify its efficacy.

### 3.4. Medication-Related Osteonecrosis of the Jaw

Medication-related osteonecrosis of the jaw (MRONJ) is a refractory disease of the maxillofacial region associated with using anti-resorptive or anti-angiogenic agents. The exact pathogenesis of MRONJ has not been fully identified, its treatment is complex, and there is no gold standard treatment. Therefore, studies applying various methods to treat MRONJ are ongoing.

The rhBMP-2 is one such attempt, but it is not applied as a single treatment but rather as an adjunctive modality combined with surgical treatment. The rationale for the application of rhBMP-2 in the treatment of MRONJ is the potential effect of its osteoinductive ability to reverse the suppression of bone remodeling in MRONJ [9]. The first study to apply rhBMP-2 to treat MRONJ was conducted by Cicciu et al. in 2012 [46]. They analyzed 20 patients with osteonecrotic lesions in the upper and lower jaws, who were treated using rhBMP-2 with a collagen carrier after removing necrotic bone without additional bone graft materials. The patients received a total dose of 4–8 mg rhBMP-2, and the follow-up period ranged from 6 to 12 months. They concluded that rhBMP-2 successfully promoted the healing of osteonecrosis of the jaws, suggesting that growth factors, particularly rhBMP-2, should be considered as therapeutic options for such patients. Since then, rhBMP-2 has been applied to defects formed after surgical treatment of MRONJ together with ACS (Figure 2) and/or platelet-rich fibrin (PRF). However, owing to the nature of MRONJ, there are no randomized controlled trials on the efficacy of rhBMP-2; instead, prospective or retrospective case-control studies are primarily conducted.

Min et al. [47] conducted a retrospective cohort study of 26 patients who underwent mandibular sequestrectomy. The experimental group (18 patients) was treated with rhBMP-2/ACS after sequestrectomy, whereas the control group (8 patients) underwent sequestrectomy without rhBMP-2/ACS. Radiographic indices were calculated using radiographs obtained immediately after surgery and more than six months later. The study found that rhBMP-2 contributed to new bone formation, with the experimental group showing a mean radiographic index of 68.4% immediately after surgery and 79.8% after six months, compared to 73.4% and 76.7% in the control group, respectively. In addition, the experimental group exhibited a significantly greater radiographic index (11.4%) increase than the control group (3.27%). These findings suggested that using rhBMP-2/ACS after sequestrectomy could be an effective treatment strategy for patients with MRONJ.

Unlike the above study, Park et al. [48] presented a prospective study comparing leukocyte-rich and PRF (L-PRF) with and without rhBMP-2/ACS applied to bone defects after surgical treatment of MRONJ. A total of 55 MRONJ patients were included, with 25 treated with L-PRF alone and 30 treated with L-PRF combined with rhBMP-2/ACS, and the surgical sites were evaluated at 4 and 16 weeks postoperatively. Patients who received both L-PRF and BMP-2 showed favorable outcomes, with complete resolution of the lesions, which was statistically significant compared to those treated with L-PRF alone. They concluded that the combined use of rhBMP-2 and L-PRF could result in the early resolution of MRONJ, making it a potentially beneficial treatment option for patients requiring ongoing anti-resorptive therapy.

There is no systematic review or meta-analysis of the efficacy of rhBMP-2 in the treatment of MRONJ, which is presumed to be difficult to perform due to heterogeneous factors such as various stages, onset sites, and induced drugs of MRONJ. Consequently, the application of rhBMP-2 in treating MRONJ is not a definitive single treatment modality; it is administered as an adjunctive treatment that can be combined with surgical treatment. Nevertheless, rhBMP-2 is considered worthy of a trial for defects formed after surgery based on its osteoinductive capacity compared to xenografts. Although no serious adverse effects have been reported, clinicians should always consider the possibility of side effects, in addition to its potential usefulness when applying rhBMP-2 to MRONJ.

### 3.5. Maxillofacial Trauma

Maxillofacial trauma includes fractures caused by traffic accidents, fighting, sports activities, etc. Since the maxillofacial region is mainly related to social and psychological aspects, treatment after trauma in this part is very important. In the field of maxillofacial trauma, efforts need to be required to utilize the advantages mentioned above of rhBMP-2, and it was found that rhBMP-2 is more effective in fracture healing by reducing the consolidation period [49]. Thus, the use of rhBMP-2 in this field is expected; however, there are no clinical studies in which rhBMP-2 has been applied in treating maxillofacial fractures. Using rhBMP-2 in maxillofacial trauma seems to be essential in restoring large defects after trauma rather than in open reduction and internal fixation.

## 4. Safety and Limitation of rhBMP-2

### 4.1. Safety of rhBMP-2

Safety is one of the most important considerations in using biomaterials, and rhBMP-2 cannot avoid such considerations. Safety issues associated with rhBMP-2 are continuously being addressed compared to other bone graft materials in maxillofacial surgery.

Several adverse events, such as ectopic bone formation [50,51], vertebral osteolysis [52,53], and postoperative radiculitis [54,55], occurred in orthopedic surgery where the application of rhBMP-2 preceded the area of maxillofacial surgery. In particular, there was a declaration by the FDA in 2008 regarding the possibility of retropharyngeal edema that could threaten the airway when rhBMP-2 is used in cervical spinal fusion [56]. Contrary to these severe complications reported in orthopedics, there is no convincing evidence of serious complications associated with using rhBMP-2 in maxillofacial surgery [57]. However, some adverse events should be noted in maxillofacial surgery, even if they are not life-threatening.

The most commonly reported rhBMP-2-associated adverse event is postoperative edema. Postoperative edema has been reported as oral and facial edema, and mouth pain and oral erythema are also commonly reported adverse events associated with using rhBMP-2 in the maxillofacial area [34]. Although these events can be resolved without unique management, clinicians must be aware of this when administering rhBMP-2 and should be vigilant about the possibility of postoperative edema, especially since the maxillofacial region is close to the airway. In addition, postoperative edema can lead to wound dehiscence, which may be problematic in terms of the effectiveness of bone grafting.

The risk of malignancy is another issue related to the safety of rhBMP-2 that must be mentioned. This is an issue since high doses of BMPs may theoretically increase cancer risk. BMPs regulate cell differentiation and growth in many tissues and can also be expressed in cancer cell lines, indicating that BMPs may enhance cancer development [58]. However, several studies have reported that using BMP is not associated with cancer risk or mortality [6,59,60]. The most recent meta-analysis of the correlation between rhBMP application and cancer incidence also found that rhBMP was not associated with an increased risk of cancer within the cohort [61]. However, when looking closely at the results of this meta-analysis, the risk ratio of the rhBMP group compared to the control group was 1.85; therefore, it cannot be said that there is no risk. In addition, in a study by Carragee et al. [62] in 2013, a representative paper on cancer risk, it was reported that 40 mg of rhBMP-2 could be associated with increased cancer risk. Fortunately, as this high dose is not applied to the maxillofacial area, the correlation between the use of rhBMP-2 and cancer risk is thought to be relatively low. To the best of our knowledge, there are no case reports or clinical studies on the occurrence of cancer associated with the application of rhBMP-2 in maxillofacial surgery. Nonetheless, the safety controversy of rhBMP-2 continues, and it is necessary to investigate the accurate identification of the correlation between dose and cancer incidence and the long-term safety in the safety mentioned above issues.

### 4.2. Limitation of rhBMP-2

In addition to safety, several other limitations make clinicians hesitant to administer rhBMP-2. First, suitable carriers or scaffolds for rhBMP-2 are lacking. The delivery system may be the most crucial factor determining the efficacy of BMP. The desired dose should be locally delivered to the target site through an appropriate carrier or scaffold to maximize the efficacy of rhBMP-2 while reducing its unwanted effects. It is known that a carrier for rhBMP-2 should be able to provide optimal cellular attachment, cellular and vascular growth, and release kinetics [63,64]. In addition, a robust carrier that can be maintained for a long time and bind to the protein is required to exert the osteoinductive effect of rhBMP-2. Although ACS does not fully satisfy these conditions, it has often been used as a carrier. ACS shows high binding ability to rhBMP-2 and has optimal biocompatibility; however, it has the major drawback of mechanical weakness. Compression of the surrounding soft tissue can result in the undesirable local release of high doses of rhBMP-2, increasing the probability of undesirable effects [65,66]. In addition, the biodegradation of ACS is challenging to predict and control, and collagen derived from bovine and porcine skin may rarely cause the production of antibodies against type I collagen [67]. Therefore, developing a carrier to replace collagen is urgent, and β-TCP, HA, and DBM have been applied as carriers in maxillofacial surgery, but a carrier surpassing ACS has not yet been developed. Therefore, it is necessary to develop a new carrier that is more robust and can be maintained for a more extended period while retaining the advantages of ACS.

The second limitation is the high cost of rhBMP-2. Although the exact cost of rhBMP-2 cannot be accurately measured as it varies by capacity and manufacturer, it is undoubtedly more expensive than other bone graft materials used in maxillofacial surgery. Mehta et al. [68] reported that higher surgical material costs were incurred when rhBMP-2/DBM was used compared to autologous iliac crest bone grafting in alveolar cleft reconstruction (rhBMP-2/DBM vs. autologous iliac crest bone graft, $1375 vs. $717). Although not a maxillofacial surgery study, McGrath et al. [69] reported that surgeons reduced the dose of rhBMP-2 during surgery when they became aware of the cost of rhBMP-2. Therefore, the high cost of rhBMP-2 is a limitation that must be overcome for further application in maxillofacial surgery.

Finally, the optimal dose of rhBMP-2 has not been determined. Since the combination of rhBMP-2/ACS was used in most studies, 1.5 mg/mL of rhBMP-2 is applied in maxillofacial surgery in most cases. This is probably related to FDA approval; however, a dose of 1.5 mg/mL cannot be considered an optimal dose of rhBMP-2. As BMPs show dose-dependent efficacy, experiments using primates have shown that lower doses are less effective regarding the quantity, quality, and duration of bone formation [70,71]. The dilemma caused by reduced efficacy due to low BMP doses and reduced safety due to high doses makes it challenging to determine the optimal dose of rhBMP-2. The investigation for an optimal dose that exhibits more efficient osteoinduction without compromising safety than the currently widely applied 1.5 mg/mL dose will have to be continued.

## 5. Conclusions

RhBMP-2 is used in various fields of the maxillofacial area to utilize its osteoinductive ability. Although rhBMP-2 is generally recognized for its efficacy in bone formation, it has not reached the gold standard as a graft material because of its high price and safety concerns. This is why clinicians are reluctant to use it compared to other bone graft materials, and as a result, rhBMP-2 is not universally used in the clinical setting of maxillofacial surgery. Continuous research is needed on subjects for which there is no consensus among researchers, such as the optimal dosing and carrier of RhBMP-2, and based on such a foundation, rhBMP-2 can overcome its limitations and become a promising graft material for maxillofacial surgery.

## Figures and Tables

**Figure 1 bioengineering-10-01005-f001:**
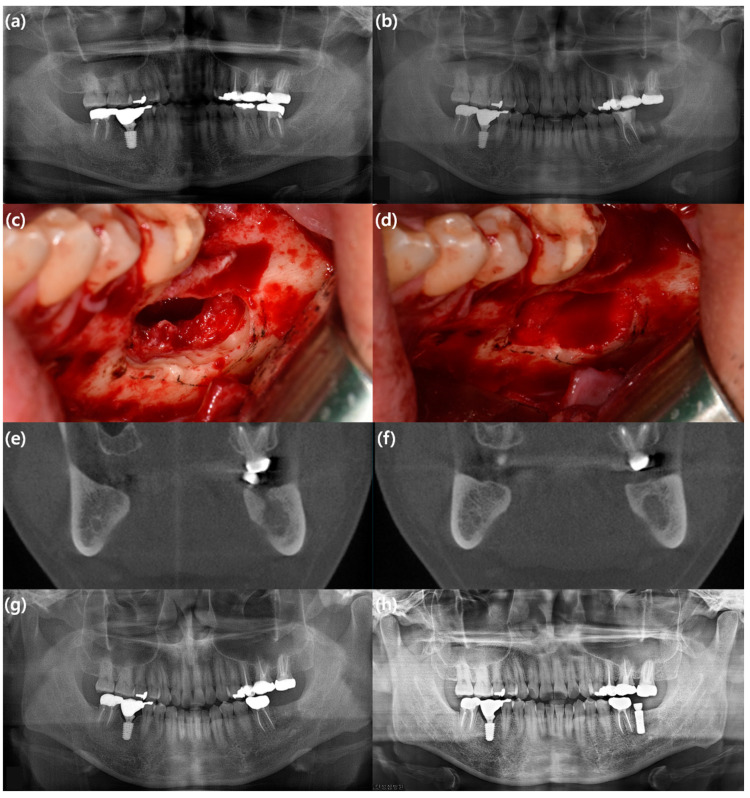
A case representing an example of alveolar ridge augmentation using rhBMP-2/ACS. A 35-year-old male patient with a cystic mass on the left mandibular posterior area underwent cyst enucleation, surgical extraction of the mandibular left second molar, and application of rhBMP-2/ACS. He showed no significant symptoms associated with the use of rhBMP-2. After uneventful healing, he underwent implant installation one year after surgery. (**a**) Preoperative panoramic view; (**b**) immediate postoperative panoramic view; (**c**) intraoperative clinical photo showing the defect after cyst enucleation and surgical extraction; (**d**) intraoperative clinical photo showing the defect filled with rhBMP-2/ACS; (**e**) preoperative coronal view of cone beam computed tomography (CBCT) image; (**f**) coronal view of CBCT image eight months after surgery; (**g**) panoramic view six months after surgery; (**h**) panoramic view one year after surgery.

**Figure 2 bioengineering-10-01005-f002:**
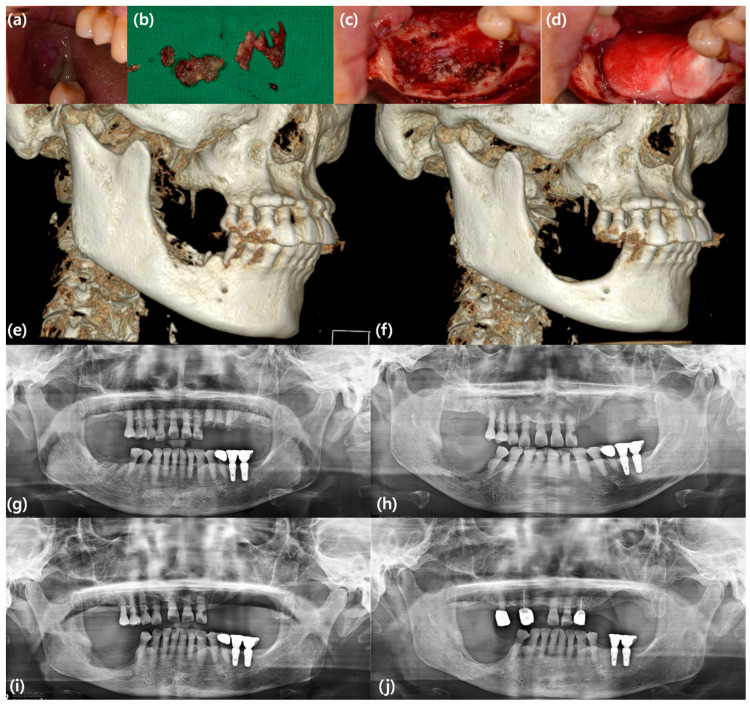
A case represents an example of MRONJ treatment using rhBMP-2/ACS. A 68-year-old female patient with MRONJ on the mandibular right posterior area underwent sequestrectomy, surgical extraction of the mandibular right second premolar, and application of rhBMP-2/ACS. She showed no significant symptoms associated with rhBMP-2 other than usual postoperative edema. Although sufficient bone regeneration was not achieved at the surgical site, she presented uneventful healing and underwent the fabrication of removable partial dentures. (**a**) Clinical photo showing denuded bone and pus discharge due to MRONJ; (**b**) clinical photo showing removed sequestrum and granulation tissue; (**c**) intraoperative clinical photo showing the defect after sequestrectomy and surgical extraction; (**d**) intraoperative clinical photo showing the defect filled with rhBMP-2/ACS; (**e**) preoperative three-dimension (3D)-reconstructed CBCT image; (**f**) 3D-reconstructed CBCT image six months after surgery; (**g**) preoperative panoramic view; (**h**) immediate postoperative panoramic view; (**i**) panoramic view three months after surgery; (**j**) panoramic view two years after surgery.

**Table 1 bioengineering-10-01005-t001:** Summary of clinical use of rhBMP-2 in maxillofacial surgery.

	Type of Combination Attempted	FDA Approval	Efficacy
Maxillary sinus augmentation	rhBMP-2/ACS rhBMP-2/Bio-Oss^®^ rhBMP-2/BCP	Approved	Confirmed
Alveolar ridge augmentation	rhBMP-2/ACS rhBMP-2/Bio-Oss^®^ rhBMP-2/β-TCP/HA rhBMP-2/DBM	Approved	Confirmed
Alveolar cleft reconstruction	rhBMP-2/ACS rhBMP-2/hydrogel rhBMP-2/DBM	Not approved (Off-label use)	Promising
MRONJ	rhBMP-2/ACS rhBMP-2/PRF	Not approved (Off-label use)	Promising
Maxillofacial trauma	Not applicable	Not approved (Off-label use)	Not confirmed

rhBMP-2, recombinant human bone morphogenetic protein-2; FDA, Food and Drug Administration; ACS, absorbable collagen sponge; BCP, biphasic calcium phosphate; β-TCP, β-tricalcium phosphate; HA, hydroxyapatite; DBM, demineralized bone matrix; MRONJ, Medication-related osteonecrosis of the jaw; PRF, platelet-rich fibrin.

## Data Availability

The data presented in this study are available on request from the corresponding author.

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
