# Peer review of "Current Status of Recombinant Human Bone Morphogenetic Protein-2 (rhBMP-2) in Maxillofacial Surgery: Should It Be Continued?"

_bioengineering, 2023, doi:10.3390/bioengineering10091005_

Round 1
Reviewer 1 Report
I appreciate the opportunity to review an article titled “Current status of recombinant human bone morphogenetic protein 2 (rhBMP-2) in maxillofacial surgery : Should it be continued?”. To date, rhBMP2 has been studied for a long time, aiming at the clinical application to bone regeneration. But I don’t think rhBMP2 has got a stable position in clinical. In such a current situation, this review has summarized the clinical applications of rhBMP2 in maxillofacial surgery that have been done so far. This report will be a good resource for us to look back at the past and to look forward to the future developments of bone tissue engineering with rhBMP2. I am going to list some points I have noticed at this time. Hope these help.
1) It would be nice to have a table summarizing the clinical use of rhBMP2 mentioned in this text.
2) Are the two cases in Figures 1 and 2 yours or someone else’s? Do you have permission to use those photos If they are someone else’s, for example, the cases shown in other reports?
3) The amount of BMPs used in clinical trials is given in concentration. I think that not only the concentration but the total amount is important in terms of adverse effects such as cancer risk. Do you know the total amount used in clinical applications?
Reviewer 2 Report
Dear authors,
This review was to examine the current status of the clinical application and efficacy of rhBMP-2 in the field of maxillofacial surgery and to review its safety and limitations, along with literature review. And the authors concluded that rhBMP-2 is a promising graft material for maxillofacial surgery, however it has not yet become the gold standard owing to safety and cost concerns. They indicated that farther research is required to establish long-term safety, optimize dosing, and develop better carriers.
I think that this review is that important because they were warning about clinical usage of rhBMP-2 for the maxillofacial surgery, and the review give the opportunity of reconsidering about use of rhBMP-2 to the both clinician and basic researchers who are involving bone metabolism.
I would like to mention the comments of the content in the paragraph of MRONJ. It is well known that rhBMP-2 is strong inducer of bone formation, and involved in the development of various organization. At the same time this mean has rh BMP-2 may have many kinds of side effect which hasn’t been clear yet. Furthermore, the detailed mechanism of MROMJ hasn’t revealed. This paper described clinical usage of MRONJ , and authors should emphasize that dangers of clinical usage of rhBMP-2 for MRONJ.
